# Implementation of the XR Rehabilitation Simulation System for the Utilization of Rehabilitation with Robotic Prosthetic Leg

Woosung Shim [1], Hoijun Kim [2], Gyubeom Lim [2], Seunghyun Lee [3], Hyojin Kim [4], Joomin Hwang [4], Eunju Lee [4], Jeongmok Cho [5], Hyunghwa Jeong [5], Changsik Pak [5], Hyunsuk Suh [5], Joonpio Hong [5,*] and Soonchul Kwon [1,*]

1 Graduate School of Smart Convergence, Kwangwoon University, Seoul 01897, Republic of Korea
2 Department of Plasma Bio Display, Kwangwoon University, Seoul 01897, Republic of Korea
3 Department of Ingenium College Liberal Arts, Kwangwoon University, Seoul 01897, Republic of Korea
4 Cybernetics Lab, Department of Plastic Surgery, Biomedical Engineering Research Center, Seoul Asan Medical Center, University of Ulsan College of Medicine, Seoul 05505, Republic of Korea
5 Department of Plastic Surgery, Seoul Asan Medical Center, University of Ulsan College of Medicine, Seoul 05505, Republic of Korea
* Correspondence: joonphong@amc.seoul.kr (J.H.); ksc0226@kw.ac.kr (S.K.)

**Abstract:** With the recent development of a digital rehabilitation system, research on the rehabilitation of amputees is accelerating. However, research on rehabilitation systems for patients with amputation of the lower extremities is insufficient. For the rehabilitation of amputees, it is important to maintain muscle mass through the improvement of muscle movement memory, continuous rehabilitation learning, and motivation to improve efficiency. The rehabilitation system in a virtual environment is convenient in that there is no restriction on time and space because rehabilitation training of amputees is possible without removing/attaching general prosthetic legs and robot prosthetic legs. In this paper, we propose an XR rehabilitation system for patients with lower extremity amputation to improve the motivational aspect of rehabilitation training. The proposed method is a system that allows patients and clinical experts to perform rehabilitation in the same environment using two XR equipment called HoloLens 2. The content was provided in the form of a game in which the number of movements of amputees was allocated as scores to enhance rehabilitation convenience and motivation aspects. The virtual 3D model prosthetic leg used in-game content worked through the acquisition and processing of the patient's actual muscle EMG (ElectroMyoGraphy) signal. In order to improve reactivity, there was a time limit for completing the operation. The classified action should be completed by the amputee within the time limit, although the number of times set as the target. To complete the operation, the amputee must force the amputation area to exceed an arbitrarily set threshold. The evaluation results were evaluated through an independent sample *t*-test. we contribute to the development of digital rehabilitation simulation systems. XR rehabilitation training techniques, operated with EMG signals obtained from actual amputation sites, contribute to the promotion of rehabilitation content in patients with amputation of the lower extremities. It is expected that this paper will improve the convenience and rehabilitation of rehabilitation training in the future.

**Keywords:** rehabilitation; EMG; lower-limb; robotic ankle

## 1. Introduction

In recent years, the cost of purchasing a prosthesis was difficult to manage and maintain due to the high price of prosthetic components [1]. However, with the grafting of a new generation of 3D printing technologies, the price of prosthetic legs has recently decreased. As the number of prosthetic users increases, rehabilitation training for patients with lower limb amputations has been developed [2]. A 3D rehabilitation method exists as an alternative to the cost of prosthetics. Several studies are being conducted in the

field of visual processing such as 2D/3D [3–5]. However, with an inexpensive prosthetic leg, when an amputee patient uses a prosthetic leg, the normal EMG signal cannot be transmitted to the prosthetic leg. In amputee patients, it is difficult to generate EMG signals due to the degeneration of muscles and nerves at the amputation site, so a prosthetic leg rehabilitation training course is required. The learning ability of the amputation robot to control the prosthetic leg was improved by developing digital rehabilitation software. The EMG signal-based digital rehabilitation simulation method was effective for therapists to quickly evaluate and systematically customize rehabilitation programs [6].

In this study, the eXtended reality rehabilitation system was developed to improve the usability of patients with lower limb amputation while wearing a robot [7,8]. eXtended reality rehabilitation application was found to be more effective than existing rehabilitation programs in terms of rehabilitation [9]. This is the reason that virtual reality and game-based therapy provide fun and motivation for rehabilitation [10,11]. A method to expand the role and use of technology has been studied through the development of an XR game-based rehabilitation program [12]. The first process of the XR rehabilitation training was acquiring EMG signals from amputee patients. The second stage was refining the raw signal data through signal processing. The final process uses the acquired data to create an interface with the Unity engine and configure the XR simulation system environment. To use this method, the rehabilitation simulation was performed using Microsoft's HoloLens2. TCP communication and virtual object interaction were used in the Unity game engine to build the HoloLens2 environment. It was a form of repeating the simulation process in multiple sessions. The software was designed to encourage participation in rehabilitation and improve treatment programs by conducting an XR system. [13,14]. We measured joint vertex movements using Azure Kinect for motion segmentation and generation.

In this paper, we list the proposed software program and the data obtained after deployment. Section 2 explains the XR Rehabilitation System, Robotic Ankle Prosthetic Leg, and related research. Section 3 describes the proposed method to conduct the experiment. Section 4 shows the data obtained by the proposed method and the usability evaluation. Finally, Section 5 presents the conclusions, limitations, and future research plans.

## 2. Background Theory

### 2.1. XR Rehabilitation Simulation System

Extended Reality (XR) includes a variety of tools that combine physical and virtual environments. This technology provides users with an immersive experience in a simulation world with which they can interact with. XR (eXtended Reality) is a collective term for technologies such as VR and MR. The XR technology is a combination of digital and physical environments. This technology brings benefits to the medical community. XR simulations are used in medical practice in situations where spatial information such as the location of medical equipment is needed [15]. XR medical rehabilitation simulations using HoloLens2 have several advantages over VR headsets and the developing process is also simple [16]. The use of 6DoF increases the sense of realism when interacting with objects.

### 2.2. Robotic Ankle Prosthetic Leg

The prosthetic leg for the lower extremity amputation patient consists of a bionic structure. The movement of the lower extremities of the amputee patient is determined with the help of the EMG sensor. The virtual prosthetic leg of the patient's amputation site was replicated and worn [17]. Amputees can use prosthetic legs to walk and perform daily activities [18]. The prosthetic leg consists of the motorized knee, ankle, and foot joints for amputee patients, and the weight and dimensions are determined based on human body data. Roll and pitch axes are applied to the ankle joint to control balance on the ground [19]. Artificial leg testing requires the analysis of data collected by IMU and EMG sensors. Testing the prosthetic leg of the amputee patient for data analysis is performed by measuring different walking conditions and walking speeds [20]. During the prosthetic gait test, GRF (Ground Reaction Force) information can be used for clinical gait analysis

and prosthesis design. Human motion can be recognized based on the data collected by the test. According to the recognition result, stable walking is possible by adjusting the control parameters of the prosthetic leg. Figure 1 below shows the structure of the robotic prosthesis.

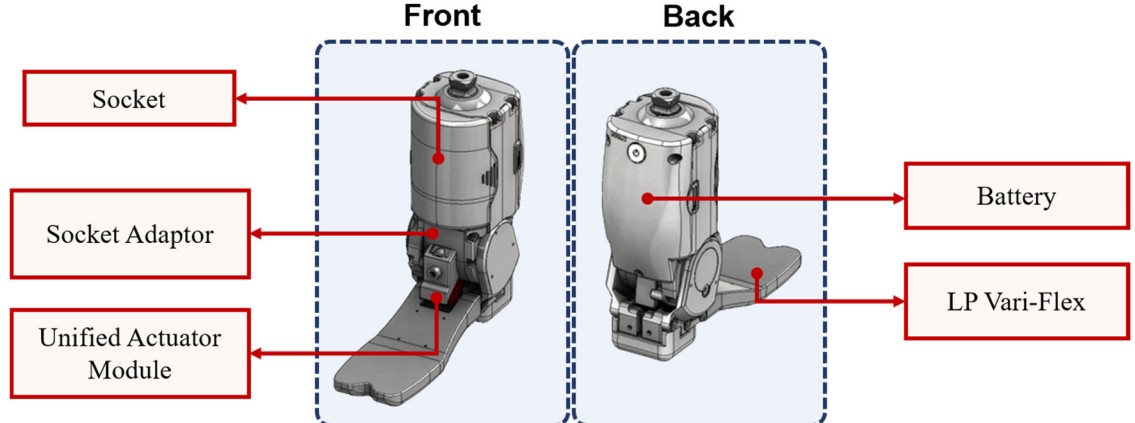

**Figure 1.** The Blueprint of the Robotic Ankle Prosthetic Leg (Socket = A part that holds the patient's amputation site. Socket Adaptor = It is a part that aligns the socket when working on the prosthesis. Unified Actuator Module = A module that performs ankle movements. Battery = A part that recharges the prosthesis. LP Vari-Flex = The LP Vari-Flex is a module for users with long cut edges that do not have enough free space for the Vari-Flex) (Source: Hugo Dynamics, Product: RoFT(gen.2), Model: HGROFT02).

*2.3. Related Work*

The amputation site is largely divided into upper and lower extremities. The lower extremities are affected more than the upper extremities. Therefore, research using virtual reality for lower extremity amputation patients is being conducted. The rehabilitation program to improve the balance and gait of patients with traumatic unilateral lower extremity amputation has been conducted in various ways (different methods such as virtual reality and augmented reality). In recent years, with the introduction of virtual reality rehabilitation, the development of methods that are fun and safe for patients has become important [21–23]. In another research using virtual rehabilitation, Azure Kinect was used to detect body movements. Infrared sensors were used to capture joints and body parts to collect data. An experiment was conducted by comparing the traditional rehabilitation program with games such as VR multidirectional motion and tennis. A virtual reality rehabilitation study showed superior efficacy in testing patients with lower extremity amputations who have sustained war injuries. The limitation is that the game is not suitable for all participants. In another rehabilitation study, there is an improved training and rehabilitation process for wearing a robotic prosthesis using virtual reality and EMG sensors.

VR Rehabilitation system was proven that is suitable for lower-extremity amputated patients. [24–26]. The rehabilitation program process of the upper extremity study is similar to the method proposed in this study. Azure Kinect is used when creating movements by measuring the range of motion of the lower leg when attaching a virtual prosthesis.

**3. Algorithm of System**

The XR rehabilitation simulation was performed using a HoloLens2, with both the amputee and the clinical expert wearing the HoloLens2 device. The patient and clinical expert wearing the device can jointly control the virtual prosthesis to be attached to the amputated site. Figure 2 below shows the XR rehabilitation simulation system.

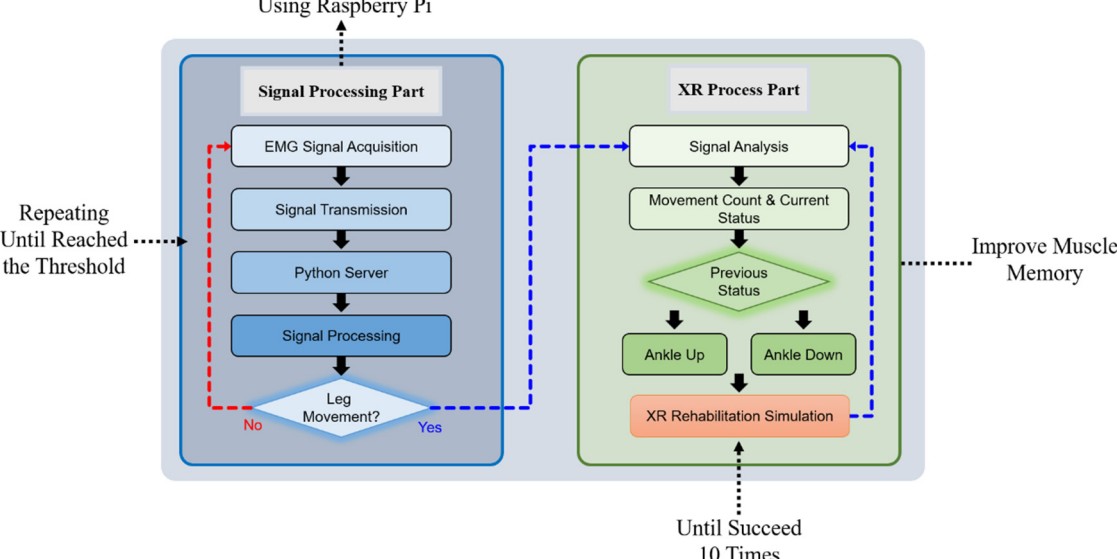

**Figure 2.** The Process of the Rehabilitation Program.

The overall system of this study was divided into two parts. It is divided into Signal Processing Part and XR Process Part, and each acts a different role. The first Signal Processing Part is responsible for obtaining and refining motion signals and handing them over to the XR Process Part. EMG Signal is obtained by attaching an electrode to the amputation part of the lower extremity amputee. The acquired signal is transmitted by wire from the Arduino Shield to the Raspberry Pi 4. Raspberry Pi 4 acts as a Python server and determines whether the movement of the lower extremity amputee is 'static' or 'dynamic' through the threshold. If it is 'static', the process proceeds again from the EMG signal acquisition end, and the behavior is not rendered in the HoloLens 2 environment. If it matches 'dynamic', the signal is passed over to the XR Process Part via TCP communication. In the XR Process Part, a count for measuring the number of times is added because there is a 'static' signal. Since there are two operations, Unity stores the previous operation state as an integer value. The current motion data is rendered according to the previous motion state, and this process is repeated by the XR rehabilitation simulation system until 10 successes.

### 3.1. XR Rehabilitation Simulation System

Figure 3 shows the architecture of the XR Rehabilitation system.

In our proposed method, the amputee patient must perform a total of 10 movements. The system is configured to allow the amputee to focus on the game by giving the 'Complete' message each time they perform a movement. During rehabilitation, the motion of the patient's joint vertex is measured using the Motion Capturing technique with Unity and Azure Kinect. The numerical value of the motion of the joint vertex of the amputee patient is measured when the actual robotic prosthetic leg is worn. For this purpose, the robot prosthetic leg was connected to a signal through Bluetooth communication. This measurement records the public's values of joint vertex motion and angles of four joints (Right Ankle, Left Ankle, Right Foot, Left Foot) required for motion in the rehabilitation simulation system. The EMG signal, the values of the joint vertex motion, and the angle data obtained by this method are stored for motion production according to the patient's EMG signal segmentation.

**XR Rehabilitation Simulation System**

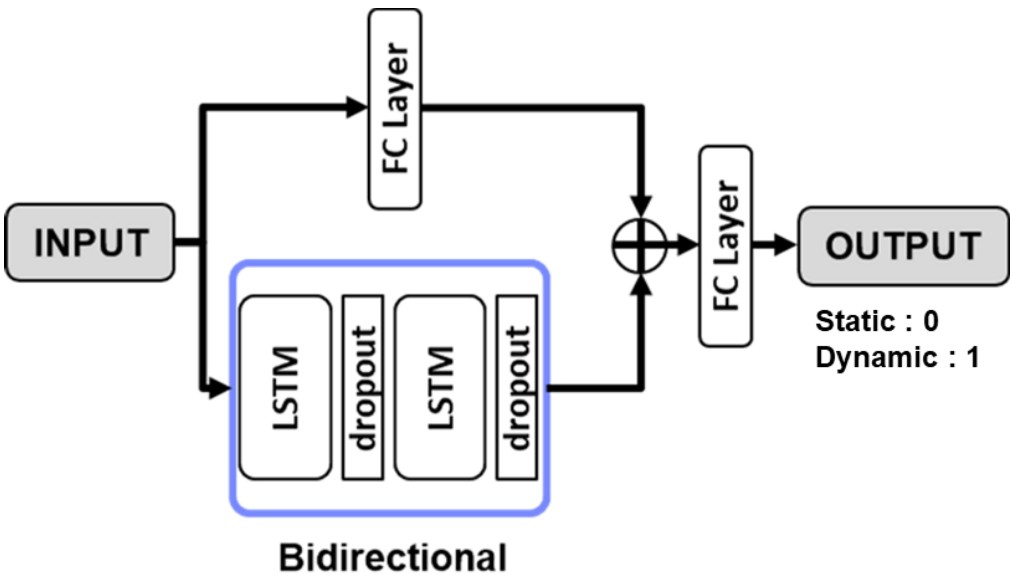

**Figure 3.** The architecture of the XR Rehabilitation System.

### 3.2. Movement Recognition

As a deep learning model, it is constructed based on an RNN model generally used in the field of signal processing. The RNN model is advantageous for processing one-dimensional data, and the CNN model can process one-dimensional and two-dimensional data but has a heavy disadvantage over the RNN model. The RNN model exhibits excellent performance in sequential and iterative data analysis. The proposed model was composed of the LSTM layer and the FC layer among the layers of the RNN model. In addition, two-way learning techniques were used to learn the reverse flow as well as the sequential flow of signals. The structure of the deep learning model is proposed in Figure 4.

**Figure 4.** Proposed Deep-Learning Model Structure.

The rehabilitation training process before wearing a robot prosthetic leg was designed as a simple ankle movement repetition process in all rehabilitation simulations. The amputee selects one of the rehabilitation simulation methods. Raise your feet in a sitting position and exercise your ankles according to the number of evaluations.

The signal processing process measures the EMG signal emitted by the amputee through the connected EMG sensor. The measured electromyogram signal is transmitted to the raspberry pie to determine whether it is a motion signal, and the obtained signal is classified into a static state (0) or a dynamic state (1). If there is the movement of leg muscles, signals are classified. Among the classified signals, a moving signal is transmitted

to the Unity game engine through TCP communication. The usability evaluation data of rehabilitation training conducted with a mixed reality rehabilitation simulation system were classified through usability evaluation and *t*-test.

For the training parameters, the optimizer was set to Adam, the epoch was set to 100, and the batch size was set to 200. The length of the data to be trained and inferred was set to 64. For the loss function, the L1 loss function is used, defined as Equation (1).

$$L_D = \sum_{i=0}^{n} |D_i - f_D(x_i)|. \tag{1}$$

Equation (1) is the Mean Absolute Error (MAE), which calculates the absolute error between the prediction result and the correct response data. $f_D$ is the detecting deep learning network, $x_i$ is the input data, and $D_i$ is the correct response data. This formula is used to calculate and learn the difference between the detected correct data and the detected deep learning prediction value.

### 3.3. Rehabilitation Content

The two basic movements of the rehabilitation software were created based on the movements that the amputee patient should perform before wearing the robotic prosthesis. When the amputee patient wears a prosthetic leg, simple repetitive movements were performed for the ankle part so that the rehabilitation training could be focused on the ankle part. Figure 5 is the point of the 3D model that serves as the virtual prosthesis of the amputee patient during the simulation. Figure 5 shows the result of measuring the movement of each major point of the lower extremity.

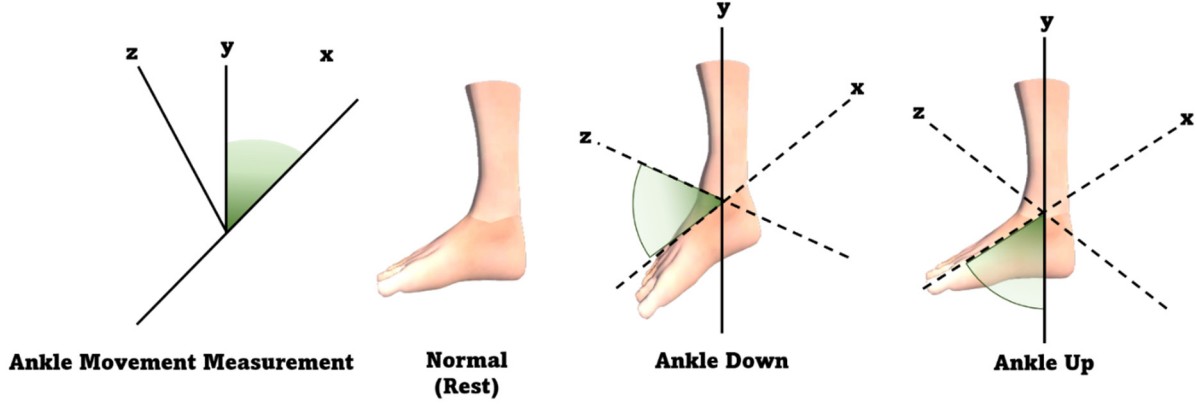

**Figure 5.** Rehabilitation Behavior Assignment.

The method proposed in this paper is an XR rehabilitation simulation system. As shown in Figure 4, the method of rigging the 3D model is based on the joint vertex motion values measured for the general public. The movement of the model is activated during transmission to the game engine based on the amputee patient's EMG signal. For the application of the proposed method, Microsoft's HoloLens2 was used to move the position value directly in the same space between the clinical expert and the patient. Interaction with objects and interactions between patients and clinical expertise are possible.

### 3.3.1. Joint Vertex Movement

To enhance the realism of the 3D model virtual prosthetic leg's animation, we used Azure Kinect to measure the joint vertex movement required for rehabilitation. Azure Kinect and Visual Studio were linked using Nuget. This was programmed with Unity's Motion Tracking and recorded 4 joints necessary for rehabilitation based on the general public. A local coordinate system of (x, y, z) was obtained from the recorded keyframe. Animation key frames were input to the main body of the 3D model virtual prosthetic leg through the measuring coordinate system. The outcome value measured in this experiment is the

standard for rehabilitative motion segmentation based on the maximum and minimum values. The standards measured in this result were used in creating actual movements. The measurement result is shown in Figure 6 below.

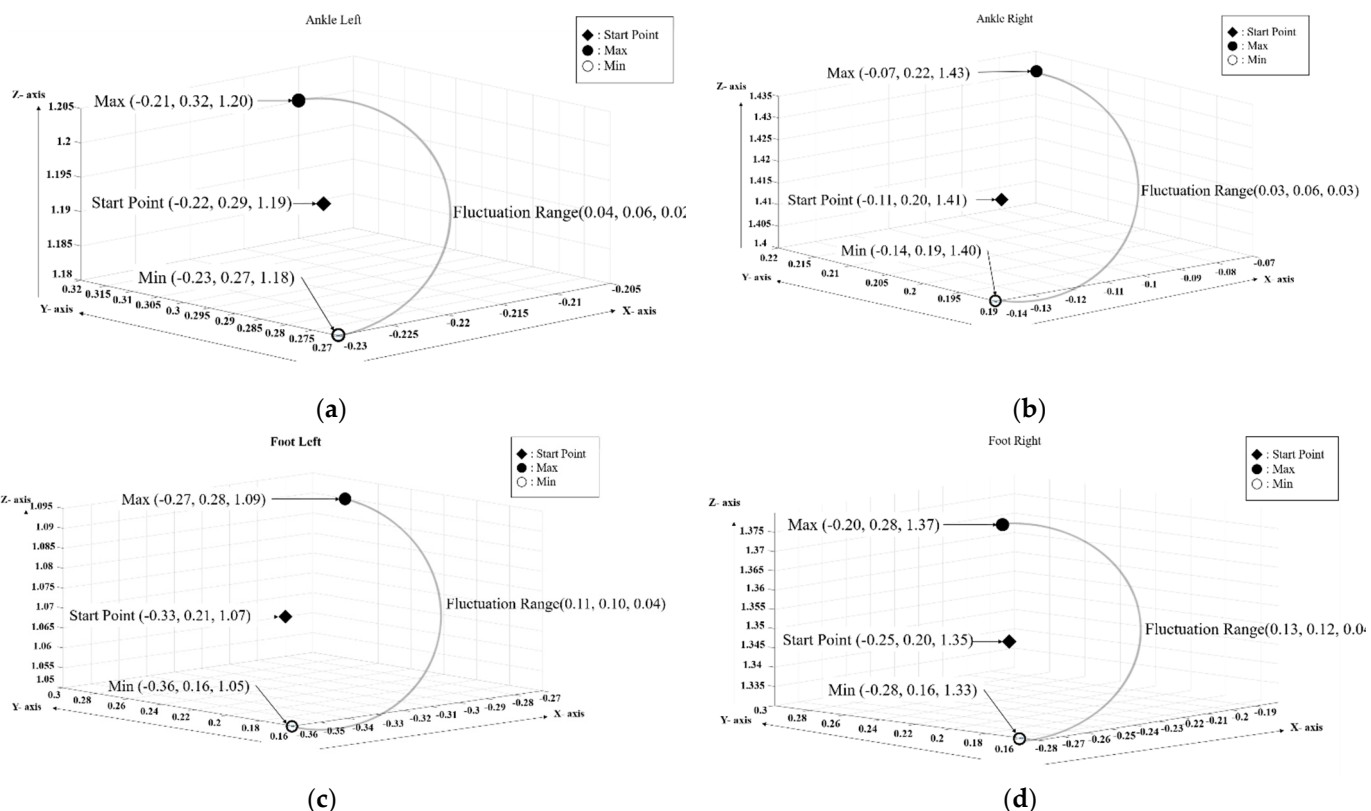

(**a**)  (**b**)

(**c**)  (**d**)

**Figure 6.** Measurement of Joint Vertex Movement ((**a**) = Ankle Left, (**b**) = Ankle Right, (**c**) = Foot Left, (**d**) = Foot Right).

During the rehabilitation simulation test, the joint vertex movement values of a total of 4 parts were measured and visualized. The outcome value measured in this experiment is the standard for rehabilitative motion segmentation based on the maximum and minimum values. The standards measured in this result were used in creating actual movements. The measurement result is shown in Figure 6. Four joint vertex movements were measured in Figure 5. (a) is Ankle Left, (b) is Ankle Right, (c) is Foot Left, (d) is Foot Right. All movement was measured in the same motion. Start Point is the coordinate for the beginning of the joint movement. The highest point of the Ankle Up motion can be confirmed by the max coordinate. Conversely, min's coordinates were calculated by measuring the minimum coordinates of the Ankle Down movement. The fluctuation range was measured by the vector operation of the maximum and minimum values of the joint movement. With the value between the fluctuations shown in this graph, it is possible to construct a more specific rehabilitation simulation system. In addition, the joint vertex movement values measured in this study were subdivided into the rehabilitation generation motions, which are evaluated based on the specificity of the signal processing values. Table 1 below lists the maximum, minimum, and variation measured values of joint vertex movement values. The unit was measured as 1 m.

**Table 1.** Coordinate Movement Figures for Each Joint.

| Joint | Max. (Unit: m) | Min. (Unit: m) | Fluctuation Range (Unit: m) |
|:---:|:---:|:---:|:---:|
| | X: −0.21 | X: −0.23 | X: 0.04 |
| Ankle Left | Y: 0.32 | Y: 0.27 | Y: 0.06 |
| | Z: 1.20 | Z: 1.18 | Z: 0.02 |
| | X: −0.07 | X: −0.14 | X: 0.03 |
| Ankle Right | Y: 0.22 | Y: 0.19 | Y: 0.06 |
| | Z: 1.43 | Z: 1.40 | Z: 0.03 |
| | X: −0.27 | X: −0.36 | X: 0.11 |
| Foot Left | Y: 0.28 | Y: 0.16 | Y: 0.10 |
| | Z: 1.09 | Z: 1.05 | Z: 0.04 |
| | X: −0.20 | X: −0.28 | X: 0.13 |
| Foot Right | Y: 0.28 | Y: 0.16 | Y: 0.12 |
| | Z: 1.37 | Z: 1.33 | Z: 0.04 |

In Table 1, the maximum and minimum values of 4 joint coordinates were measured. The motions used for measurement were the Ankle Up and Ankle Down motions used in the XR rehabilitation simulation system. Ankle Left, the first measurement joint of the subject in a seated condition from Azure Kinect, was 126 cm away. The maximum coordinate value when performing the Ankle Up motion was vector 3 (−0.21, 0.32, 1.20), and the minimum coordinate value when performing the Ankle Down motion was measured as vector 3 (−0.23, 0.27, 1.18). This resulted in the fluctuation range of the fluctuation range (0.04, 0.06, 0.02). In the same posture, position, and motion, the subject's Ankle Right was at a distance of 146 cm. The maximum coordinate values were vector 3 (−0.07, 0.22, 1.43), and the minimum coordinate values were vector 3 (−0.14, 0.19, 1.40). The fluctuation range (0.03, 0.06, 0.03) was changed.

The distance between the subject's third measurement joint, Foot Left, and Azure Kinect was 125 cm. The maximum coordinate values were vector 3 (−0.27, 0.28, 1.09), and the minimum coordinate values were vector 3 (−0.36, 0.16, 1.05). Fluctuation ranges (0.11, 0.10, 0.04) were measured. The last joint, Foot Right, was located at a distance of 145 cm from Azure Kinect. The maximum coordinate value is vector 3 (−0.20, 0.28, 1.37), and the minimum coordinate value is vector 3 (−0.28, 0.16, 1.33). The magnitude of change in this joint was measured using the fluctuation range (0.13, 0.12, 0.04).

### 3.3.2. 3D Model 6DoF Acquisition

HoloLens2 is placed in the XR environment by automatically setting up an image processing 6DoF Unity. The 6DoF actual application formula of the 3D model for the user is as follows. First, Equation (2) is the rotation value of 6DoF. The equations for pitch, roll, and yaw are matrix values and are defined as the rotation matrix of the *x*-axis. The *y*-axis in Equation (3) rotation and the *z*-axis in Equation (4) are defined as follows:

$$R_x(\psi) = \begin{bmatrix} 1 & 0 & 0 \\ 0 & cos\psi & -sin\psi \\ 0 & sin\psi & cos\psi \end{bmatrix}, \tag{2}$$

$$R_y(\theta) = \begin{bmatrix} cos\theta & 0 & sin\theta \\ 0 & 1 & 0 \\ -sin\theta & 0 & cos\theta \end{bmatrix}, \tag{3}$$

$$R_z(\phi) = \begin{bmatrix} cos\phi & -sin\phi & 0 \\ sin\phi & cos\phi & 0 \\ 0 & 0 & 1 \end{bmatrix}, \tag{4}$$

$\psi$, $\theta$ and $\phi$ are Euler angles, and by combining these equations, the coordinate movement amount of rotation, which is one of the 6 DoF coordinates, can be obtained. The matrix shown in Equation (5) can be considered as the order of three rotations about each

circular axis. It first rotates around the *x*-axis of the model's local coordinates, and then around the *y*-axis. When rotating around the *z*-axis, the rotation then appears as a series of rotation matrix products.

$$R = R_z(\phi)R_y(\theta)R_x(\psi)$$
$$= \begin{bmatrix} cos\theta cos\phi & sin\psi sin\theta cos\phi - cos\psi sin\phi & cos\psi sin\theta cos\phi + sin\psi sin\phi \\ cos\theta sin\phi & sin\psi sin\theta sin\phi + cos\psi cos\phi & cos\psi sin\theta sin\phi - sin\psi cos\phi \\ -sin\theta & sin\psi cos\theta & cos\psi cos\theta \end{bmatrix} \tag{5}$$

Given a rotation matrix R, equalize each element and the corresponding element of the matrix product $R_z(\phi)$, $R_y(\theta)$ and $R_x(\psi)$ to calculate the Euler angle, $\phi$, $\theta$, and $\psi$ to determine the position of the prosthetic leg. This formula was used to find the Euler angle and determine the virtual prosthetic position instantiated in Hololens2.

The rehabilitation training process before wearing the robotic prosthesis was designed as a simple ankle movement repetition process for all rehabilitation simulations. The amputee selects one of the rehabilitation simulation methods. Raise the feet in a sitting position and perform ankle exercises corresponding to the number of assessments. During the exercise, the patient attaches the EMG sensor to the amputation of the lower extremity and then continues with the movement. In addition, we measure joint vertex movement counts through Azure Kinect for the general public. The numerical values are subdivided and standardized for the rehabilitation of motion animation and motion classification. Figure 7 below shows the rehabilitation process.

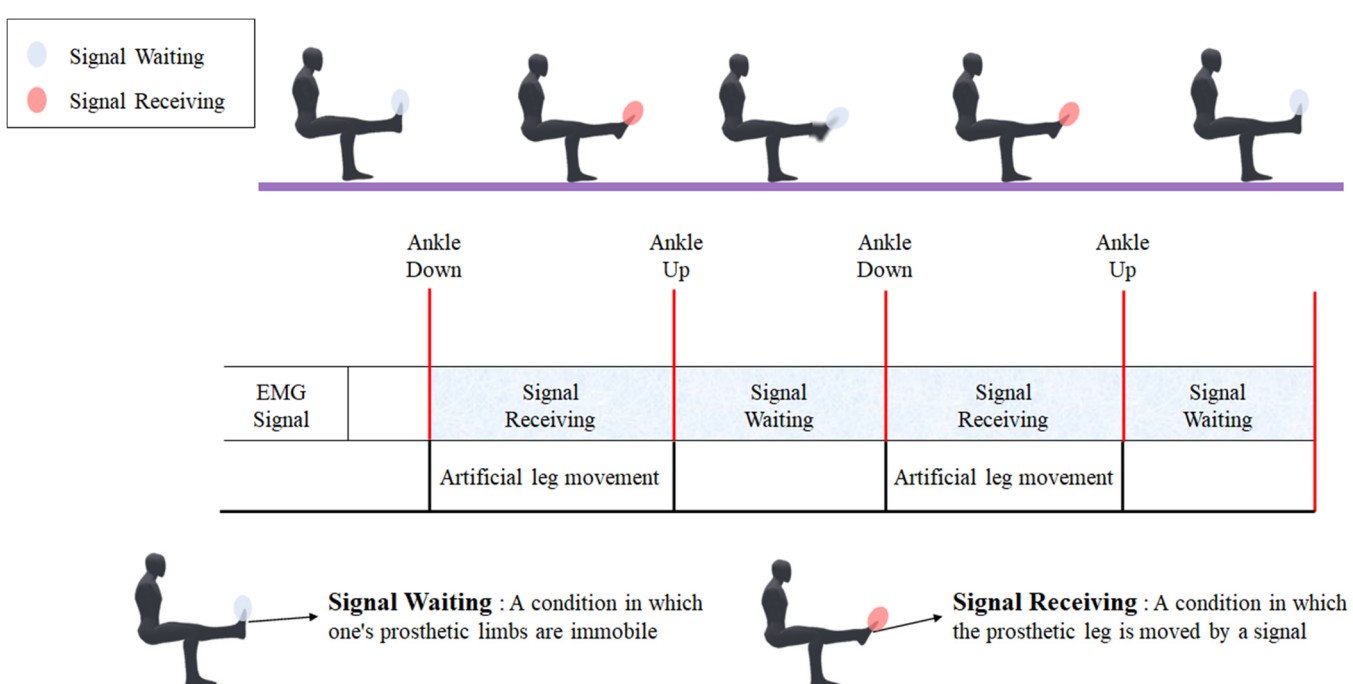

**Figure 7.** Rehabilitation Process for Patients with Lower Extremity Amputation.

The movements of the amputee patient shown in Figure 7 consist of 'ankle up' and 'ankle down'. During the simulation, the signal obtained from the EMG sensor is transmitted to the Raspberry Pi, and signal processing continues. Since the EMG signal generates a lot of noise during acquisition, it is used to interpret the signal as motion or no motion. The captured EMG signal is transmitted to the Unity 3D Rendering Engine. Based on this signal, the simulation moves the 3D model according to the signal as shown in Figure 7. Signal receiving represents the dynamic motion in the presence of a signal. Signal waiting indicates the static state of the 3D model when no motion signal is detected. Therefore, in the figure, if the foot part of the 3D model is red, it is moving because a signal is present,

and if it is blue, it is stationary because no signal is present. Based on the above movements and algorithms, an XR rehabilitation system was developed. Through the proposed motion, the test was conducted after the EMG signal was connected to actual amputee patients and the general public. In the case of an upper limb amputated patient, there are various classifications of actions that can be performed. Lower-limb amputated patient has a few classifications method 'Ankle Up' and 'Ankle Down'. The recognition of the EMG signal was proceeded by threshold.

### 3.4. Rehabilitation Method

The configuration of the XR Rehabilitation Software system in Unity and the virtual prosthetic thread wearing according to 6DoF are shown in Figure 8.

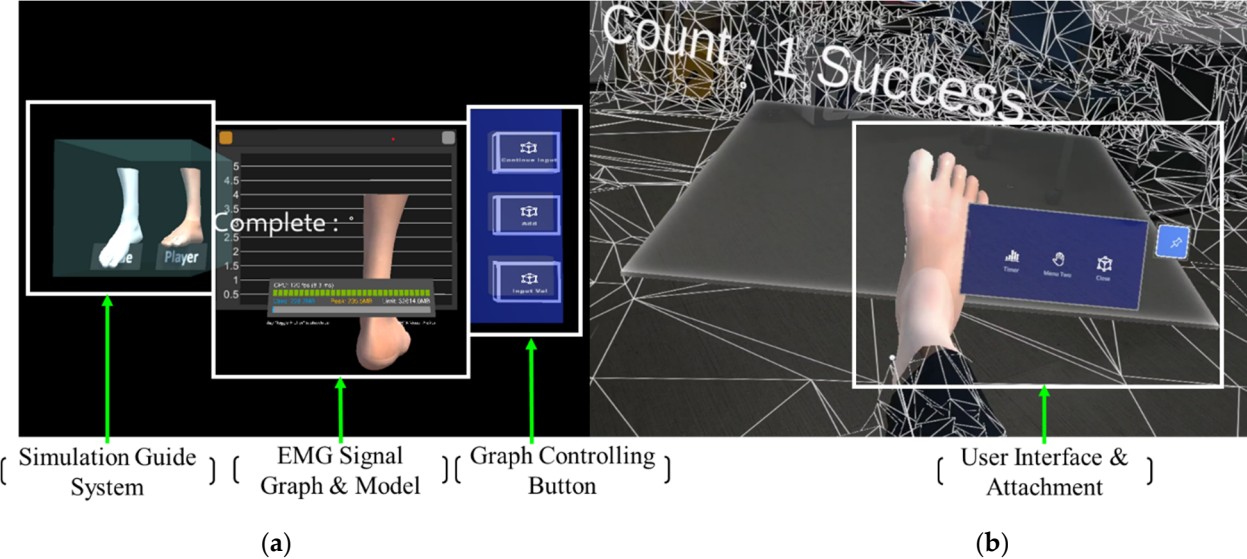

(**a**)　　　　　　　　　　　　　　　　　　　　　　　　　　　　(**b**)

**Figure 8.** Composition and Actual Operation of XR System ((**a**) = Full Screen of Simulation Program Component, (**b**) = Actual Operating Figure).

Figure 8a shows a virtual prosthetic leg worn by the patient and models that provide a guide to the patient. In addition, the EMG graphic extracted from the patient's leg can be reviewed on HoloLens 2. There is also a button to check the graph by channel. Figure 8b the simulation operation was performed normally 10 times by the repeated operation signal. Figure 9 is an image of the actual amputation patient.

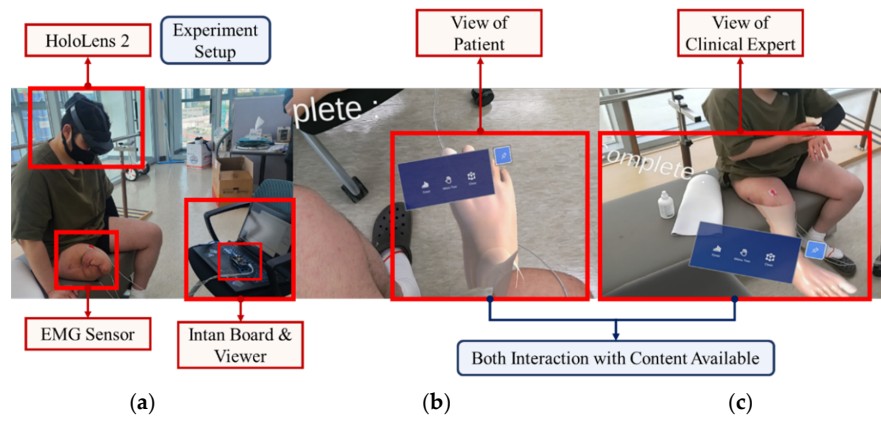

(**a**)　　　　　　　　　　　　(**b**)　　　　　　　　　　　　(**c**)

**Figure 9.** The Configuration of the Amputation Patient Actual Test Environment and the Viewpoint of the Participants in the Multi-Session ((**a**) = Experiment Setup, (**b**) = Patient's Viewpoint, (**c**) = Clinical Expert's Viewpoint).

Figure 9 shows a photo of an amputation patient and a clinical expert wearing the HoloLens2 at the same time. (a) shows a picture of a real amputated patient wearing HoloLens2 with an EMG sensor attached. (b) shows the field of view of the amputated patient. (c) shows the view of an expert. Both the expert and amputee tested the interaction in one session.

## 4. Results

### 4.1. Experimental Environment

In this section, the environment used for the proposed method is presented. In this virtual environment, C# and Unity 3D Rendering Engine were used for language and software. In the XR rehabilitation simulation, a multi-environment using two or more HoloLens2 devices was configured. In the extended reality environment, 6DoF coordinate axes were used. Communication between the Python server and the Unity Client uses the TCP protocol. Data measurement was performed using Azure Kinect and Intan RHD 2216. Both devices were used to capture and visualize lower extremity motion figures and EMG signals used in rehabilitation software. Table 2 below shows the experimental environment.

**Table 2.** Experimental Environment.

| Category | eXtended Reality |
|:---:|:---:|
| Device | HoloLens 2 |
| | Raspberry Pi 4 Model B |
| Measuring Device | Azure Kinect |
| | Intan RHD 2216 |
| Language | C# |
| Software | Unity (2020.3.8) |
| Number of Axes | 6 Degrees of Freedom |
| OS | Windows Holographic |

The Signal Processing section measures the EMG signal from the amputated sited and emitted by the amputee patient via the connected an EMG sensor and 3 electrodes. The measured EMG signal is transmitted to the Raspberry Pi to determine whether it is a motion signal or not, and the signal is classified as 0 or 1. The communication between RHD 2216 Arduino Shield and Raspberry pi 4 is serial communication using USB b to a. If there is a movement of the leg muscles, the signal is processed. Signals with motion among the classified signals are transmitted to the Unity 3D Rendering Engine via TCP communication. The configuration of the device to acquire the EMG signal is as follows. It consists of Intan's RHD2216 Arduino Shield, Arduino Uno, and Raspberry PI4. The EMG signal was acquired at the amputated site of the amputee patient and recorded with 1 channel of the surface electromyography signal. The inputs to the on-chip ADC is 0.10 V−2.45 V Range. In the acquisition system, the number of samples per time is 1 kHz and the resolution is ±4.4 peak to the peak. A 60 Hz notch filter was used to remove power noise during signal acquisition. In the low pass filter, the butterworth filter is used and the cut off at 5 Hz. The high pass filter is cut off at 75 Hz. Signals with a weaker intensity than the threshold are classified as 'static'. Equation (6) is an equation of a threshold value set for each patient with amputation of the lower extremity.

$$Threshold = \sqrt{\frac{\sum(x_i - \bar{x})^2}{n}} \times r, \; r = 20 \tag{6}$$

The threshold value is set by multiplying the standard deviation value of the raw EMG signal average by 20. The number of parameter inputs per layer in the Bidirectional LSTM model is 108,422. Real-time prediction of raw signals up to 65,000 Hz per second is possible. The hyperparameters of the deep learning model used in the experiment are shown in Table 3.

**Table 3.** Network Hyperparameter.

| Parameter | Value |
|---|---|
| Epoch | 100 |
| Btach Size | 200 |
| Optimizer | Adam |
| Initial Learning Rate | 0.005 |
| Scheduler | Polynomial Decay |
| Dropout | 0.5 |

For the training parameters, the optimizer was set to Adam, epoch to 100, and batch size to 200. The length of the data to be trained and inferred was set to 64. The initial learning rate was 0.005, and the scheduler used Polynomial Decay. The dropout is 0.5. The hardware specifications and deep learning frameworks used for learning the deep learning model are shown in Table 4.

**Table 4.** Hardware Specification and Deep Learning Framework.

| | |
|---|---|
| CPU | AMD 5800x |
| Ram | 32 GB |
| GPU | NVIDIA RTX 3090 24 GB |
| Framework | Tensorflow 2.6.0 |

In this experiment, the dataset used for learning, verification, and evaluation was conducted 100 times with 5 normal groups and 5 cut groups who received test consent from the mixed reality rehabilitation simulation system. Data acquisition was done for a total of 4 s with 2 s of 'static state' and 2 s of 'dynamic state'. The number of data used for learning and evaluation of the deep learning model is shown in Table 4.

### 4.2. Evaluation Method

4.2.1. EMG Signal Motion Recognition

Accuracy was used to evaluate the results of EMG learning. The accuracy equation used to proceed with the evaluation is shown in (7).

$$Accuracy = \frac{TP + TN}{TP + TN + FP + FN}. \tag{7}$$

$TP + TN + FP + FN$ is the sum of the number of EMG data samples obtained from the cut area of the lower extremity amputees. $TP + TN$ represents the number of detections of static and dynamic signals.

4.2.2. *t*-test Analysis

To evaluate the XR rehabilitation simulation system proposed in this paper, a heuristic evaluation method was used to test the software program users. Heuristic evaluation is an evaluation method that provides accurate results with a small number of participants. The system was evaluated by a total of 15 users. 8 patients were from the general public, and 7 patients were amputees. The age range of the evaluators was from 30 to 60 years old, and different age groups participated. The evaluators had given informed consent to the study, and they were familiar with the experiment. There is a total of 21 evaluation items. During the evaluation, participants were asked about the suitability of the software. The scores were required to be given on a scale of 1 to 10, with a lower score indicating non-conformity.

The *t*-test is a statistical analysis technique that compares the means between two groups. The *t*-test is a method to verify the statistical significance of the mean difference between two groups. The formula for the t-test is as follows.

$$T = \frac{\Delta X - \mu}{\frac{S}{\sqrt{N}}}. \tag{8}$$

$\Delta X$ is the mean of the difference between the two groups, $\mu$ is the mean of the population, and S is the standard deviation of the difference between the two groups. The significance level ($\alpha$) was set to 0.05. When *p*-Value < ($\alpha$), it indicates that the difference between the means is statistically significant. In this paper, two *t*-tests were analyzed. The first was divided by amputation status, and the second was divided by age. Table 5 shows the items of a usability test.

**Table 5.** Items of Usability Test.

| Number | Items |
| --- | --- |
| 1.1 | It has an extroverted shape considering the amputation patient |
| 1.2 | It was an interface for patients with amputated lower limbs |
| 2.1 | The physical controls were convenient to reach |
| 2.2 | The operation method was simple |
| 2.3 | There was a help function for users. |
| 2.4 | It was easy to switch to another menu when needed. |
| 2.5 | It responded to the user's different usage environments (home, hospital, etc.) |
| 2.6 | It was easy to access functions frequently. |
| 3.1 | Rehabilitation time was shortened compared to conventional methods. |
| 3.2 | The rehabilitation training process was more effective than the existing methods. |
| 3.3 | Compared with the existing method, the virtual prosthesis helped to accurately perform rehabilitation training. |
| 3.4 | Rehabilitation training was more efficient than existing methods. |
| 3.5 | It was easy to adapt to the Hololens2 device. |
| 3.6 | It was easy to adapt to the virtual prosthesis rehabilitation training. |
| 3.7 | Convenience compared with the existing rehabilitation methods. |
| 3.8 | There was a motivating factor compared with the existing rehabilitation method. |
| 3.9 | The virtual prosthesis was more friendly and comfortable than the robotic prosthesis. |
| 3.10 | Overall satisfied with this program |
| 3.11 | The rehabilitation content of this program was more helpful than the existing methods. |
| 3.12 | Whether this program can be reused. |
| 3.13 | I want to recommend this program to other amputated patients. |

*4.3. Results of EMG Signal Processing and* t-*test*

4.3.1. EMG Signal Processing Results

Figure 10 shows the amputee patient and the extracted signal graph.

The *x*-axis is time(sec) and the *y*-axis is amplitude($\mu$V). The number of parameters of the model is 108,422, and the inference time is 0.98 ms. Real-time predictions are possible up to about 65,000 Hz, and the inference accuracy is 92.43% was shown.

The second is a model for detecting whether an EMG signal is dynamic or static. Figure 11 is a graph of the loss and accuracy of the proposed Bidirectional LSTM.

Figure 11a shows the learning accuracy, Figure 11b shows the learning loss rate. Table 6 shows the accuracy and loss of the deep learning model.

**Table 6.** Accuracy & Loss of Deep Learning Model.

| | |
| --- | --- |
| Accuracy | 92.43% |
| Loss | 0.405% |
| Inference Time | 0.98 ms |

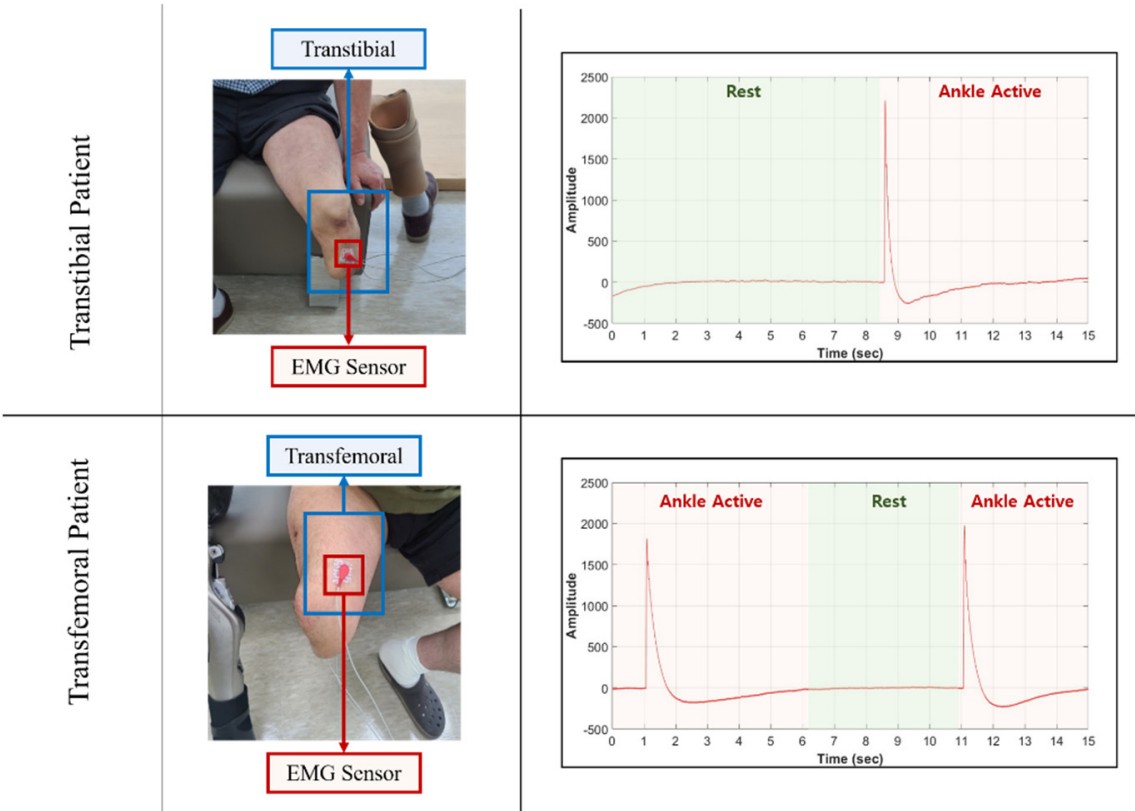

**Figure 10.** EMG Sensor Attachment and Extracted EMG Signal Graph.

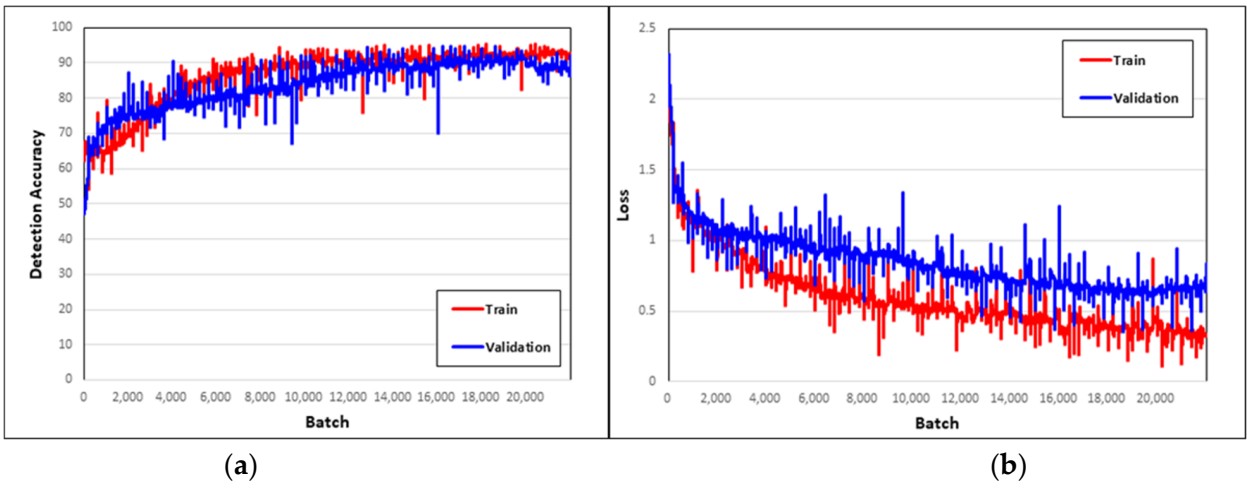

(**a**)　　　　　　　　　　　　　　　　　　　　　　　　　　　(**b**)

**Figure 11.** Loss and Accuracy of the Proposed RNN Model. (**a**) Accuracy of the proposed RNN Model (**b**) Loss of the proposed RNN Model.

The result when using epoch 100 and batch size 200. The accuracy of motion presence detection was 92.43%. The inference time is 0.98 m, and real-time prediction is possible up to about 65,000 Hz.

### 4.3.2. *t*-test Results

Table 7 and Figure 12 below shows the result of the evaluation and independent *t*-test result.

**Table 7.** Evaluation Results and Scores by Amputated Status.

| Serial | Items | Public Group (N = 8) M ± SD | Amputation Group (N = 7) M ± SD | z | *p*-Value |
|---|---|---|---|---|---|
| 1 | 1.1 | 6.375 ± 1.847 | 7.000 ± 1.291 | −0.652 | 0.514 |
|   | 1.2 | 6.000 ± 2.138 | 6.571 ± 2.300 | −0.409 | 0.682 |
| 2 | 2.1 | 6.375 ± 2.386 | 6.143 ± 1.952 | −0.409 | 0.682 |
|   | 2.2 | 6.875 ± 1.959 | 7.000 ± 2.000 | −0.237 | 0.812 |
|   | 2.3 | 6.000 ± 2.563 | 6.714 ± 0.951 | −0.299 | 0.765 |
|   | 2.4 | 6.375 ± 2.133 | 5.714 ± 1.380 | −1.063 | 0.288 |
|   | 2.5 | 7.625 ± 2.200 | 5.143 ± 1.070 | −2.245 | *0.025 |
|   | 2.6 | 7.000 ± 2.070 | 4.857 ± 1.215 | −2.102 | *0.036 |
| 3 | 3.1 | 7.000 ± 2.138 | 5.857 ± 1.573 | −1.425 | 0.154 |
|   | 3.2 | 7.000 ± 2.138 | 5.857 ± 1.773 | −1.251 | 0.211 |
|   | 3.3 | 7.125 ± 2.800 | 6.581 ± 1.512 | −1.001 | 0.317 |
|   | 3.4 | 7.000 ± 2.138 | 5.857 ± 1.864 | −1.254 | 0.210 |
|   | 3.5. | 4.625 ± 1.408 | 6.571 ± 1.272 | −2.294 | *0.022 |
|   | 3.6 | 6.750 ± 2.435 | 5.000 ± 1.633 | −1.756 | 0.079 |
|   | 3.7 | 7.375 ± 2.387 | 5.428 ± 1.397 | −2.051 | *0.040 |
|   | 3.8 | 8.000 ± 2.777 | 6.857 ± 1.345 | −1.704 | 0.088 |
|   | 3.9 | 7.125 ± 2.700 | 7.714 ± 2.360 | −0.480 | 0.631 |
|   | 3.10 | 7.125 ± 2.232 | 7.286 ± 1.254 | −0.303 | 0.762 |
|   | 3.11. | 7.375 ± 1.922 | 7.000 ± 1.155 | −1.027 | 0.305 |
|   | 3.12 | 7.250 ± 2.605 | 7.429 ± 1.512 | −0.236 | 0.813 |
|   | 3.13 | 7.000 ± 2.828 | 8.000 ± 1.732 | −0.648 | 0.517 |

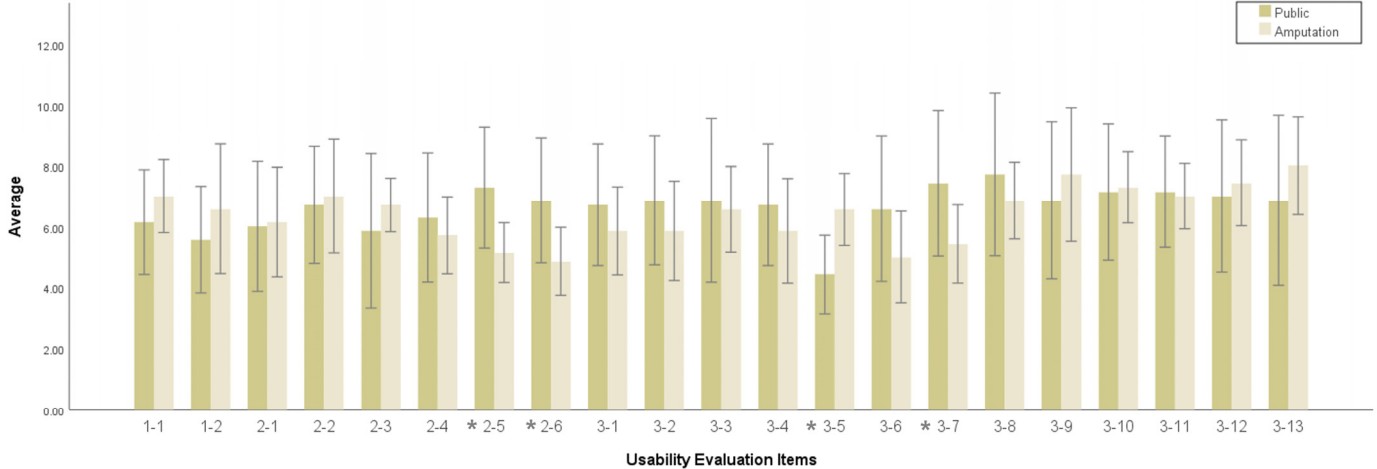

**Figure 12.** Evaluation Results and Scores by Amputated Status (*: *p* < 0.05).

Serial No. 1 is acceptance evaluation, No. 2 is usability evaluation, and No. 3 is reactivity and aesthetic evaluation. In the evaluation index, most users had difficulty 2-5getting used to using the HoloLens2.

In the comparison of the normal group (4.625 ± 1.408) and amputation group (6.571 ± 1.272), the normal group was high, and the evaluation comparison in various rehabilitation environments was significant (z = −2.245, * *p* = 0.025). In the comparison between the normal group (7.000 ± 2.070) and the amputation group (4.857 ± 1.215) of the 2–6 questions, the evaluation of the convenience of function use was significant (z = −2.102, * *p* = 0.036). The comparison between the normal group (4.625 ± 1.408) and the amputation group (6.571 ± 1.272) in the 3–5 questions showed high evaluation of the amputation group, and the comparison of the adaptive part evaluation using HoloLens 2 showed significance (z = −2.294, * *p* = 0.022). The comparison of the normal group (7.375 ± 2.387) and the amputation group (5.428 ± 1.397) in the 3–7 questions showed high evaluation of the

normal group, and significance in the evaluation comparison of existing rehabilitation methods (z = −2.051, * $p$ = 0.040).

Older users had trouble adjusting to HoloLens2 and gave it a low score for needing a tutorial to control the rehabilitation system. Lower-aged users gave high ratings for reactivity and gaming ability. In both age groups, it was the motivational aspect that scored high in the detailed survey and evaluation. Rehabilitation training in a virtual environment was rated as motivating and fun in the evaluation process.

In the usability evaluation of this program, the ease of use was judged from an expert's point of view, and the reactivity of the system was highly rated. However, when evaluated by amputee patients and the general population, the HoloLens2 adaptation method received a very low rating. Users provided feedback on the voice guidance system, the improvement of the attachment effect, and the addition of a number of 3D models. Among amputee patients, it was felt that a lot of practice on the HoloLens2 was needed before using the program. The overall usability rating has a strong motivating factor for simulating a virtual environment, but there were difficulties in using and customizing Hololesn2.

## 5. Discussion

In recent years, the virtualization of amputation patient rehabilitation simulation has been advanced through the development of digital software. This study provided an XR environment technology. Robotic prosthetic limbs have lower training repeatability and a lower sense of accomplishment due to weight and comfort issues compared to virtual worlds where rehabilitation training can be easily trained. The method presented in this study enables interaction with virtual prostheses in real space. The interactive prosthetic leg can be comfortably attached and sized to reduce the rehabilitation preparation process. The structure and joint vertex motion values of the XR rehabilitation simulation system obtained in this study were well-evaluated. In the part of the rehabilitation technique presented in this study, it was created based on the presence or absence of motion signals. Currently, rehabilitation training for XR amputation patients expressed discomfort about the use of XR equipment. Furthermore, the development of electromyography signal acquisition technology in the medical field is needed to improve the rehabilitation system. In the future, it is expected that the measurement criteria for the segmentation of rehabilitation motion will vary according to the input force. Additionally, as the size of XR devices shrinks and supply chains have been established, a series of studies on amputation patients is expected.

## 6. Conclusions

We proposed the XR rehabilitation simulation system for lower extremity amputation patients to enhance rehabilitation convenience. The proposed system acquired EMG signals from amputated patients and proceeded in the XR environment. The motion was created by recording the maximum/minimum of ankle movement using Azure Kinect based on the general public. EMG signals were classified as 0 and 1 and passed to unity, and actions were executed in real-time according to the signals. Patients and clinical experts use HoloLens 2 in a remote location and perform rehabilitation in the same space. Spatial Anchor sharing is designed for this system. When the test was conducted on lower extremity amputation patients, it was highly rated in terms of muscle rehabilitation effect and motivation. In this study, there was a limitation in that the accuracy of the rehabilitation movement was lowered by dividing the movement into two signals. An elderly patient with lower extremity amputation expressed difficulties in using the HoloLens2 device. In future research, the motion should be subdivided according to the value of the input force to compensate for the limitations shown in the experiment. In order to provide a simplified and optimized rehabilitation simulation system, the difficulty of the user interface should be reduced. The proposed rehabilitation system is expected to be used by clinical professionals and amputees in actual wards. This system is currently conducting a clinical trial in an Asan medical hospital. In future studies, this research has to set the threshold according to the EMG signal suitable for each individual. It is expected that the simulation system will

help improve muscle movement memory and the convenience of rehabilitation training in lower extremity amputation patients.

**Author Contributions:** Conceptualization, W.S., H.K. (Hoijun Kim), H.K. (Hyojin Kim), J.H. (Joomin Hwang) and S.K.; methodology, W.S.; software, W.S. and H.K. (Hoijun Kim); validation, W.S., G.L., E.L., J.C. and H.J.; formal analysis, W.S. and H.K. (Hoijun Kim); investigation, G.L.; resources, H.K. (Hoijun Kim) and G.L. and S.L.; data curation, W.S.; writing—original draft preparation, W.S. and S.K; visualization, W.S. and H.K. (Hoijun Kim); supervision, C.P., H.S., J.H. (Joonpio Hong), S.K and S.L.; project administration, C.P., H.S., J.H. (Joonpio Hong), S.K., S.L.; All authors have read and agreed to the published version of the manuscript.

**Funding:** This research received no external funding.

**Institutional Review Board Statement:** Not applicable.

**Informed Consent Statement:** Not applicable.

**Data Availability Statement:** The Data Presented in this Study are available on request from the corresponding author. The data are not publicly available due to privacy reason.

**Acknowledgments:** The present research has been conducted by the Research Grant of Kwangwoon University in 2022. This research was supported by the MSIT (Ministry of Science and ICT), Korea, under the ICAN (ICT Challenge and Advanced Network of HRD) program (IITP-2022-RS-2022-00156215) supervised by the IITP (Institute of Information and Communications Technology Planning & Evaluation).

**Conflicts of Interest:** The authors declare no conflict of interest.

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
