# Peer review of "Implementation of the XR Rehabilitation Simulation System for the Utilization of Rehabilitation with Robotic Prosthetic Leg"

_applsci, doi:10.3390/app122412659_

Round 1

Reviewer 1 Report

This paper proposes a new rehabilitation system using virtual reality and augmented reality. Indeed, an XR rehabilitation system for lower extremity amputation patients was proposed. The virtual prosthetic leg is operated through the patient's muscle EMG signal acquisition and processing. An usability evaluation was conducted after experimenting with the system for patients as well as the general public. 

This is an interresting work having a good impact on the rehabilitation field, however, I have some comments and questions:

1. Abstract should be optimized illustrating the main contribution.

2. Section3:  First paragraph explains the principle of the emg acquisition and processing. it should be rewritten describing the main architecture of the proposel system.

3. Section 3.1: Acquisition system lacks some details including the number of samples per time, the resolution... in addition, the selection of the notch filter and the sampling frequency are not justified. Authors should justify also the communication between the Arduino and the Raspberry is missed. 

4. The use of the kinect sensor should be justified. 

Concluding, the paper has potential to be appreciated by the readers and the above comment are formulated such that to enhance its impact.

Author Response

Dear Reviewer

Thank you for inviting us to submit a revised draft of our manuscript entitled, Implementation of the XR Rehabilitation Simulation System for the Utilization of Rehabilitation with Robotic Prosthetic Leg to Applied Science. We also appreciate the time and effort you have dedicated to providing insightful feedback on ways to strengthen our paper. Thus, it is with great pleasure that we resubmit our article for further consideration. We have incorporated changes that reflect the detailed suggestions you have graciously provided. We also hope that our edits and the responses we provide below satisfactorily address all the issues and concerns you have noted.

To facilitate your review of our revisions, the following is a point-by-point response to the questions and comments delivered in your letter dated 05 / Dec / 2022.

Point 1: Abstract should be optimized illustrating the main contribution.

Response 1: We agree with you and have amended this suggestion throughout our paper. The results defined in this paper have been added to the abstract section. We have added contributions that develop from our results and can proceed to future digital rehabilitation simulation systems (p. 1, lines 19~41).

Point 2: Section3: The First paragraph explains the principle of emg acquisition and processing. it should be rewritten to describe the main architecture of the proposel system.

Response 2: This is a valid assessment of our explanation and Figure 2. We rewrite it to describe the main architecture of our system (p. 4~5, lines 165~179).

Point 3: Section 3.1: The Acquisition system lacks some details including the number of samples per time, and the resolution... in addition, the selection of the notch filter and the sampling frequency are not justified. Authors should justify also that the communication between the Arduino and the Raspberry is missed.

Response 3: We have reflected on this comment by justifying the number of samples per time and resolution. The Notch filter and the sampling frequency are added either (p.12, lines 380~384). We have clarified that the communication between the Arduino and Raspberry pi throughout the paper (p. 12, lines 371~372).

Point 4: The use of the Azure Kinect sensor should be justified.

Response 4: You have raised an important point. We have incorporated your comments by justifying the reason and method of using Azure Kinect (p. 6, lines 235~241).

P.S
We describe the motion recognition method by adding Movement Recognition to 3.2. We added an artificial intelligence signal processing environment to the experimental environment and added formulas and descriptions for threshold setting. We describe the method for the experiment by adding the Evaluation Method to organize the results. We added Validation to the figure in Figure 11. We added Table 6 to describe the artificial intelligence learning results.

Thank you for giving us the great opportunity to strengthen our manuscript with your valuable comments and queries. We focused to update our paper depending on your important feedback. I hope that these revisions are fully satisfied to you
Thank you for your effort

Woosung Shim

Reviewer 2 Report

Implementation of the XR Rehabilitation Simulation System for the Utilization of Rehabilitation with Robotic Prosthetic Leg

The manuscript presents a rehabilitation system by means of virtual reality for people with lower extremity amputation. The idea is very interesting, and the procedure followed very good explain.

The work shows an interesting point of view, and I would like to ask you some questions:

-        How was EMG signal acquired? How many sensors and electrodes have you used in the experiment?

-        In the test it is said that “in amputee patients, it is difficult to generate EMG signals due to the degeneration of muscles and nerves”. How have you guarantee to have a good signal? Could this procedure be used for any person?

-        It is not clear how the movement is detected. Have you defined some kind of pattern?

-        It is explained that a 60 Hz notch filter have been used to remove noise. Using a raspberry, Arduino to get EMG signal could introduce too much noise to your signal. How have you considered your noise level was appropriated for your system?

-        What kind of ADC have been used to get EMG data?

-        Figure 3. What units are the y-axis?

-        Line 170. How many tests do you need to train the tool?

Author Response

Dear Reviewer

Thank you for inviting us to submit a revised draft of our manuscript entitled, Implementation of the XR Rehabilitation Simulation System for the Utilization of Rehabilitation with Robotic Prosthetic Leg to Applied Science. We also appreciate the time and effort you have dedicated to providing insightful feedback on ways to strengthen our paper. Thus, it is with great pleasure that we resubmit our article for further consideration. We have incorporated changes that reflect the detailed suggestions you have graciously provided. We also hope that our edits and the responses we provide below satisfactorily address all the issues and concerns you have noted.

To facilitate your review of our revisions, the following is a point-by-point response to the questions and comments delivered in your letter dated 05 / Dec / 2022.

Point 1: How was EMG signal acquired? How many sensors and electrodes have you used in the experiment? 

Response 1: We have revised the text to reflect your queries. Electrodes were attatched to the amputees of the lower-limb amputated patients. The patient was asked to follow the guide motion rendered in the XR rehabilitation simulation system, and an input EMG signal was acquired. We used RHD 2216 Arduino Shield Sensor connected with 3 electrodes(+, -) to acquire data in this experiment (p. 12, lines 368~369). 

Point 2: In the test it is said that “in amputee patients, it is difficult to generate EMG signals due to the degeneration of muscles and nerves”. How have you guarantee to have a good signal? Could this procedure be used for any person?

Response 2: That is an important query. We are currently conducting a clinical trial of an XR rehabilitation simulation system for lower-limb amputated patients at Asan Medical Hospital in Seoul, South Korea. EMG signals in patients with patients are weak, and we are using amplifiers to strengthen the signal. In future studies, we plan to set the threshold according to the EMG signal suitable for each individual. If the EMG signal is set according to the individual EMG signal may be guaranteed. Subsequent studies attempt to demonstrate the effectiveness of this signal. It can be applied to everyone if clinical trials are conducted on a large number of patients in hospital (p.18, lines 539~542).

Point 3:  It is not clear how the movement is detected. Have you defined some kind of pattern?

Response 3: You have raised an important question. In the case of an upper limb amputated patient, there are various classifications of actions that can be performed. However, in the case of lower-limb amputated patients, there were few classifications of movements. So, the pattern recognition proceeded by threshold. We recognized the behavior by setting thresholds that could distinguish between ‘static’ and ‘dynamic’ states. We have revised the paragraph to reflect your comments (p. 10, lines 327~332, p. 12, lines 387~390).

Point 4:  It is explained that a 60 Hz notch filter have been used to remove noise. Using a raspberry, Arduino to get EMG signal could introduce too much noise to your signal. How have you considered your noise level was appropriated for your system?

Response 4: You have raised important questions. This is a valid assessment of removing noise. We are actually using a different EMG system. The noise level using the system is also ±10µV and the noise level using the RHD 2216 Arduino Shield is under ±10µV. There was no problem in obtaining EMG signals due to the small noise range. In addition, when the EMG signals were determined to be ‘Dynamic’, the threshold value was greater than noise. Also, µV has a small noise level range, there was no problem in acquiring an EMG dataset.

Point 5:  What kind of ADC have been used to get EMG data?

Response 5: We have added a new sentence of ADC. We used RHD 2216 Arduino Shield. In addition, the inputs to the on-chip ADC is the 0.10V-2.45V Range (p. 12, lines 378~379).

Point 6:  Figure 3. What units are the y-axis?

Response 6: We have included an additional explanation of the y-axis. The units of the y-axis is µV (p. 15, lines 449~450).

Point 7: Line 170. How many tests do you need to train the tool?

Response 7: We test 100 epochs to train the tool. Overfitting occurred when learning was conducted over 100 epochs. We have refelected this questions by adding a Table 3. Network Hyperparameter (p. 13, Table 3).

P.S
We describe the motion recognition method by adding Movement Recognition to 3.2. We added an artificial intelligence signal processing environment to the experimental environment and added formulas and descriptions for threshold setting. We describe the method for the experiment by adding the Evaluation Method to organize the results. We added Validation to the figure in Figure 11. We added Table 6 to describe the artificial intelligence learning results.

Thank you for giving us the great opportunity to strengthen our manuscript with your valuable comments and queries. We focused to update our paper depending on your important feedback. I hope that these revisions are fully satisfied to you
Thank you for your effort

Woosung Shim
